# Sperm Selection and Embryo Development: A Comparison of the Density Gradient Centrifugation and Microfluidic Chip Sperm Preparation Methods in Patients with Astheno-Teratozoospermia

**DOI:** 10.3390/life11090933

**Published:** 2021-09-07

**Authors:** Cagla Guler, Sureyya Melil, Umit Ozekici, Yaprak Donmez Cakil, Belgin Selam, Mehmet Cincik

**Affiliations:** 1Institute of Health Sciences, Maltepe University, 34857 Istanbul, Turkey; cagla.guler@medicalpark.com.tr (C.G.); mehmet.cincik@maltepe.edu.tr (M.C.); 2IVF Clinic, Maltepe University Hospital, 34844 Istanbul, Turkey; sureyya.melil@medicalpark.com.tr (S.M.); umit.ozekici@maltepe.edu.tr (U.O.); 3Department of Obstetrics and Gynecology, Faculty of Medicine, Maltepe University, 34844 Istanbul, Turkey; 4Department of Histology and Embryology, Faculty of Medicine, Maltepe University, 34857 Istanbul, Turkey; yaprak.cakil@maltepe.edu.tr; 5Department of Obstetrics and Gynecology, School of Medicine, Acibadem Mehmet Ali Aydinlar University, Unit of ART, Acibadem Altunizade Hospital, 34662 Istanbul, Turkey

**Keywords:** male infertility, astheno-teratozoospermia, microchip, density gradient centrifugation, embryo, blastocyst

## Abstract

In recent years, microfluidic chip-based sperm sorting has emerged as an alternative tool to centrifugation-based conventional techniques for in vitro fertilization. This prospective study aims to compare the effects of density gradient centrifugation and microfluidic chip sperm preparation methods on embryo development in patient populations with astheno-teratozoospermia. In the study, the semen samples of the patients were divided into two groups for preparation with either the microfluidic or density gradient methods. Selected spermatozoa were then used to fertilize mature sibling oocytes and the semen parameters and embryo development on days 3 and 5 were assessed. While the density gradient group was associated with a higher sperm concentration, motility (progressive and total) was significantly higher in the microfluidic chip group. No significant differences were observed in the fertilization rates or grade 1 (G1) and grade 2 (G2) proportions of the third-day embryos. Furthermore, while the proportions of the poor, fair and good blastocysts on day 5 did not differ significantly, excellent blastocysts (indicating high-quality embryos) were observed in a significantly higher proportion of the microfluidic chip group. When compared to the classical density gradient method, the microfluidic chip sperm preparation yielded sperm with higher motility and higher quality blastocysts at day 5; in patients with astheno-teratozoospermia.

## 1. Introduction

Male factors alone account for 20–30% of infertility cases, with another 20% stemming from both male and female factors [1]. With increasing popularity since its introduction in the early 1990s, intracytoplasmic sperm injection (ICSI) became the ideal approach to treat severe male factor infertility. ICSI involves the injection of a single spermatozoon into the oocyte cytoplasm, enabling conception and pregnancy with suboptimal semen quality [2]. Though ICSI is a powerful tool, total fertilization failure still takes place in 1 to 3% of ICSI cycles [3].

A successful cycle requires the identification of the highest quality of the spermatozoa [4]. Sperm defects have been shown to negatively impact embryo quality and development [5,6,7], emphasizing the critical importance of sperm selection. While through the oviduct spermatozoa with the highest fertilization capability and the best potential for supporting embryo development have the ability to achieve fertilization; the selection of the most competent cells remains a challenge in vitro [8]. Density gradient centrifugation (DGC) [9], a classical sperm preparation method, allows for separation based on motility, size and density as the spermatozoa are centrifuged through a colloidal silica gradient. Although it is a common procedure in ICSI laboratories, semen samples with low sperm content, high viscosity and a large percentage of cellular debris are not suitable for the DGC procedure [10]. Moreover, several studies have pointed to increased DNA fragmentation in spermatozoa undergoing DGC [11,12,13,14]. Sufficient evidence has been provided by meta-analyses to show the detrimental effect of DNA fragmentation on clinical pregnancy following IVF and/or ICSI [15,16,17]. Microfluidic chip-based sperm sorting is a recent and powerful tool, which has been introduced as an alternative to centrifugation-based, conventional techniques [18,19,20]. A variety of microfluidic devices with different approaches ranging from passive to flow or chemical-based sorting have been designed to efficiently isolate the highly motile and healthy sperm [21,22,23,24,25,26]. These technologies aim to develop a reliable and accurate system that enables high-throughput, functional sperm sorting similar to the natural sperm selection process [27]. The microfluidic technique was shown to be associated with reduced DNA damage and fewer reactive oxygen species, thus improving the selection of sperm for use with ICSI [28,29]. A recent study reported a significantly diminished proportion of sperm with double-stranded DNA fragmentation [30]. Moreover, the microfluidic sorting method yielded highly motile spermatozoa with the ability to maintain membrane integrity and mitochondrial function related to ATP production in bovine spermatozoa [31]. The microchip-based technique proposed by Anbari et al. was shown to provide significantly higher progressive motility, a fraction of Class I sperm morphology and decreased DNA fragmentation. The researchers also found improved clinical outcomes with increased rates of high-quality embryo, implantation and pregnancy [32].

To our knowledge, there is a lack of data in the literature about the effects of microfluidic chip-based sperm sorting for astheno-teratozoospermia on embryo development. In the present study, we included patients with astheno-teratozoospermia, characterized by reduced sperm motility and abnormal morphology, and compared the effects of DGC and microfluidic chip-based sperm preparation by employing a commercially available microfluidic device (Fertile Ultimate^®^) on subsequent embryo development.

## 2. Materials and Methods

### 2.1. Patient Selection

Twenty-two couples who applied to Maltepe University Hospital ART Center between January 2020–September 2020 and met the study criteria were included in this prospective study. The women ranged in age from 18 to 39 and had at least two mature oocytes. A pre-implantation genetic diagnosis was not performed and MI oocytes were excluded from the study. Astheno-teratozoospermia was described according to the WHO 2010 guidelines and Kruger’s strict criteria (sperm progressive motility less than 32% and sperm morphology less than 4%). Split semen samples from the same population of infertile men with astheno-teratozoospermia were used to fertilize the mature sibling oocytes from the same women. Baseline characteristics of the patients are depicted in Appendix A. Simple randomization using a closed envelop method was used to randomize the sibling oocytes. A CONSORT 2010 Flow Diagram is provided in Appendix A. This study was approved by the Ethics Board of Maltepe University, Istanbul with the protocol number 2019/07-14.

### 2.2. Ovarian Stimulation

The standard ovarian stimulation consisted of pituitary downregulation either by GnRHa leuprolide acetate (Lucrin 0.5 mg/mL, Abbott, Madrid, Spain) or GnRH antagonist cetrorelix acetate (Cetrotide, Baxter Oncology GmbH, Halle, Germany). GnRHa was injected daily during the late luteal phase before starting the treatment cycle. The GnRH antagonist was injected daily by the 5th day of the treatment cycle. Both injections were sustained until the trigger of ovulation. Baseline ultrasounds of the patients were performed and ovarian cysts >2 cm were ruled out before starting the IVF cycle. Gonadotropins were started on cycle days 2 or 3. The daily dosages were individualized between 150 and 300 IU. All patients were monitored regularly by ultrasound until three follicles with maximum diameter >17 mm were observed. HCG 10000 U (Choriomon, IBSA, Lodi, Italy) and 5000 U hCG (Choriomon, IBSA, Lodi, Italy) plus 0.2 mg triptorelin acetate (Gonapeptyl, Ferring GmbH, Kiel, Germany) were used as the trigger for oocyte maturation in the agonist and antagonist cycles, respectively. Approximately 35–36 h after ovulation was triggered, a transvaginal ultrasound-guided oocyte retrieval was performed, under general anesthesia, with a 17-gauge needle.

### 2.3. Semen Analysis

Semen samples were collected by masturbation, after 2–7 days of sexual abstinence, from male partners with astheno-teratozoospermia. Semen analysis was performed following a period of incubation for 15–60 min for liquefaction. Ten microliters of each sample were dropped on a Makler counting chamber (Sefi-Medical Instruments, Haifa, Israel) and evaluated under a phase-contrast microscope for sperm count and motility. The semen smear was prepared and stained with Spermac™ for the assessment of sperm morphology.

### 2.4. Sperm Preparation Using the Density Gradient Centrifugation Method

Each semen sample was divided into two for a comparison of the two sperm selection methods. The first half, a liquefied semen sample for DGC, was gently layered on top of a 50:90% PureSperm density gradient containing colloidal silica particles (Nidacon, Mölndal, Sweden) in a 15 mL conical tube. Following centrifugation at 300× *g* for 20 min, the recovered pellet was washed in the PureSperm wash medium. After a second centrifugation at 500× *g* for 10 min, the pellet containing the selected motile population was re-suspended in a sperm culture medium and used in further steps.

### 2.5. Sperm Preparation Using a Microfluidic Sorting Chip

The second halves of the semen samples were prepared using the Fertile Ultimate^®^ (Koek Biotechnology, Izmir, Turkey) microfluidic sperm sorting chips. Following liquefaction, 3 mL of unprocessed semen samples were introduced to the inlet of the microchip. Next, 1.8 mL of sperm-washing solution was added to the outlet with a sterile syringe. After 30 min of incubation at 37 °C, the liquid containing the most motile and functional subpopulation accumulated in the upper-exit chamber (outlet) and was drawn with a sterile syringe.

### 2.6. Embryo Development

Following sperm preparation from the split samples by either method, selected spermatozoon was used to fertilize one of the mature sibling oocytes. The numbers of sibling oocytes for each woman and the sperm preparation method are depicted in Appendix A. ICSI was performed when the oocyte-corona complexes were denuded and were incubated for 2 h. Embryo development and blastocyst formation were evaluated on days 1, 3 and 5 based on the criteria reported by Veeck and Zaninovic [33]. Blastocysts were classified as poor, fair, good, or excellent based on the grading system by Gardner and Schoolcraft [34].

### 2.7. Statistics

Statistical evaluation was performed with the R Stats Package (R Foundation for Statistical Computing, Vienna, Austria) and the data was examined with the Shapiro Wilk test for normality. Mann Whitney U-test from non-parametric analyzes was used for data without normality assumptions in pairwise comparison analyses and independent samples *t*-test was used for analyses with normality assumptions. A *p*-value of less than 0.05 was accepted as statistically significant.

## 3. Results

The average age of males and females in the study were 37.27 ± 5.16 and 32.55 ± 4.68, respectively. The demographic data of 22 semen samples, including sperm characteristics such as the semen volume (mL), sperm concentration (10^6^/mL), total motility (%) and progressive motility (%) and morphology, are demonstrated in Table 1.

The semen samples of 22 patients with astheno-teratozoospermia were divided into two groups and each half was prepared with either the microfluidic chip (microchip group) or DGC (gradient group). Semen parameters including semen volume (mL), sperm concentration (10^6^/mL), total motility (%) and progressive motility (%), after sperm preparation with either method, are given in Table 2. While the sperm concentration was significantly higher in the gradient group (19.80 ± 19.90 vs. 4.37 ± 6.05; *p* < 0001), sperm motility was higher in the microchip group (progressive motility: 31.73 ± 19.90 vs. 68.41 ± 24.57; *p* < 0001 and total motility: 53.27 ± 24.32 vs. 79.50 ± 17.19; *p* < 0004).

ICSI was performed on a total of 203 MII oocytes and 186 of them were successfully fertilized. Subsequent embryo development was compared between the microchip and gradient groups. Table 3 shows the proportion of embryo development by sperm selection methods. Representative images illustrate the grading and classification of the developing embryos (Figure 1). As depicted in Table 3, no significant differences were observed in the fertilization rates or grade 1 (G1) and grade 2 (G2) proportions of the third-day embryos. Moreover, the proportions of the poor, fair and good blastocysts on day 5 did not differ significantly between the study groups. However, the proportion of the excellent blastocysts on day 5 was significantly higher when the spermatozoa were selected using the microfluidic chip rather than DGC (0.42 ± 0.28 vs. 0.23 ± 0.23; *p* = 0.029), indicating the presence of more high-quality embryos in the microchip group.

## 4. Discussion

In recent years, microfluidic chip-based sperm sorting has emerged as an alternative tool to historical sperm preparation methods [18,19,20]. The microfluidic application allows for rapid isolation of poor semen samples with high motility, improved DNA integrity and low morphological abnormalities [35]. The basic idea of sperm sorting in microchips is to more closely replicate in vivo physiological conditions to improve sperm selection and increase the possibility of achieving a successful ICSI outcome [27].

In this study, the spermatozoa from the split semen samples were prepared using either microfluidic platforms or DGC. Mature sibling oocytes were subjected to ICSI with sperm selected from either preparation method and the subsequent embryo development was monitored. DGC yielded a significantly higher sperm concentration, while both progressive and total motility were found higher when the Fertile Ultimate^®^ microchip system was employed in accordance with related literature [36,37,38]. A higher sperm concentration, yet lower motility in the gradient group, indicates the presence of both motile and immotile spermatozoa in the sample after DGC. On the other hand, the sperm sorted using the microfluidic chip led to significantly higher progressive and total motility, which is particularly important in our study population with astheno-teratozoospermia. Reduced progressive motility together with poor sperm morphology was shown to be associated with impaired blastocyst development and diminished quality after ICSI [39]. Particularly, due to the negative impacts on fertilization rate [40,41], decreased motility was suggested as a predictive marker for compromised fertilization after ICSI [42].

Although no difference was observed in fertilization rate, we found a significantly higher proportion of the excellent blastocysts on day 5 in the microchip group compared to the gradient group. The system used to grade blastocyst formation was introduced by Gardner and Schoolcraft in 1999 and provides a guideline in classifying the degree of blastocyst expansion, the morphological appearance of the inner cell mass (ICM) and trophectoderm cells [43]. Accordingly, the excellent blastocysts on day 5 shared the ideal features attributed to day 5 blastocysts with high viability; including an expended blastocoel cavity, well-formed ICM and trophectoderm cells, and zona pullucida thinning.

The selection of the embryo with the highest implantation potential is fundamental for a successful pregnancy [44]. In addition to increased sperm motility, by mimicking the physiological environment, microfluidic sorting also improved morphology, and caused less DNA damage in comparison to swim-up and DGC methods [10,30,32]. Further studies compared the microfluidic chip or DGC methods in terms of DNA integrity and reported a significantly lower sperm DNA fragmentation rate in microchip groups [28,38,45]. It is important to note that the DNA fragmentation index correlates negatively with embryo quality and pregnancy outcome [46]. Also, fewer reactive oxygen species were detected in sperm prepared using the microfluidic chip in comparison to those prepared by DGC or swim-up methods [29,37]. However, a recent study did not demonstrate any differences in fertilization rate, embryo quality, blastocyst development, or pregnancy rate after sperm selection with the microfluidic sorting or swim-up method [47]. On the other hand, another study conducted in 2021 reported increased rates of high-quality embryo, implantation and pregnancy by microfluidic sperm sorting [32]. Additionally, a randomized controlled trial involving 181 patients with male factor infertility reported an enhanced pregnancy rate in the microfluidic group than that in the gradient group when the female age is above 35 and the total motile sperm count ranges between 1 and 5 million [48].

Microfluidic systems were also investigated for the isolation of spermatozoa from testicular specimens of non-obstructive azoospermic men [49]. Microfluidics and nanotechnology are further explored for sperm sex-sorting before IVF to prevent sex-linked genetic diseases [50]. Besides all the aforementioned improvements, the microfluidic chip method also has the benefit of being a simple and time-saving sperm selection tool with reduced sample volumes. Conversely, despite the continued evolvements in the technology, the system is not yet widely adopted in routine laboratory practice due to the higher costs in comparison to the centrifugation-based conventional techniques. The major limitations of our study include the low patient number and the absence of blinding. However, to our knowledge, this is the first study with higher embryo numbers evaluating embryo development with sperm prepared by either DGC or microfluidic chip methods. Additional research is required to explore the pregnancy and delivery rates. The results might provide preliminary data for a larger study or a starting point for the initiation of sperm selection by the microfluidic chip method in those patients with astheno-teratozoospermia.

## 5. Conclusions

Our study included patients with astheno-teratozoospermia, who would benefit from the advantages of the MFC technology. In a clinical setting, when compared to the classical DGC method, the microchip platform yielded sperm with higher motility and higher quality blastocysts at day 5. Further studies with an increased number of patients and with more applications may provide comprehensive data for the clinical outcomes.

## Figures and Tables

**Figure 1 life-11-00933-f001:**
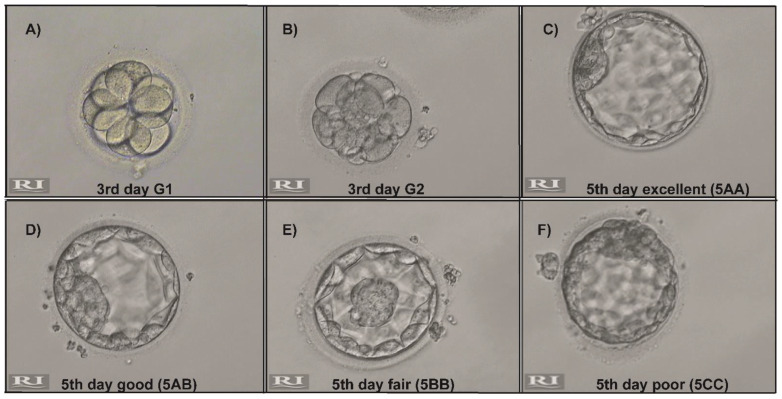
Morphological grading demonstrating (**A**) 3rd day grade 1 (G1) (**B**) 3rd day grade (G2) (**C**) 5th day excellent (5AA) (**D**) 5th day good (5AB) (**E**) 5th day fair (5BB) and (**F**) 5th day poor (5CC) embryos. The images were provided by Prof. Mehmet Cıncık.

**Table 1 life-11-00933-t001:** Demographic data of 22 semen samples. Data are shown as mean ±SD.

	Basal
Semen volume (mL)	3.18 ± 1.48
Sperm concentration (10^6^/mL)	40.63 ± 42.00
Total motility (%)	33.14 ± 14.47
Progressive motility (%)	12.59 ± 8.33
Morphology	1.50 ± 0.67

**Table 2 life-11-00933-t002:** Comparison of semen parameters after sperm preparation with a microfluidic chip or DGC methods.

	Microchip Group (n: 22)		Gradient Group (n: 22)		z/t	*p* Value
	Mean ± SD	Median (Q_1_–Q_3_)	Mean ± SD	Median (Q_1_–Q_3_)		
Volume of recovered sperm (mL)	1.07 ± 0.15	1.00 (1.00–1.05)	0.73 ± 0.08	0.70 (0.70–0.80)	−5.914 ^z^	*p* < 0.0001 *
Sperm concentration (10^6^/mL)	4.37 ± 6.05	2.15 (0.70–5.25)	19.80 ± 19.90	14.50 (6.25–28.00)	−3.911 ^z^	*p* < 0.0001 *
Total motility (%)	79.50 ± 17.19	81.00 (70.00–92.25)	53.27 ± 24.32	55.00 (33.00–73.50)	−3.547 ^z^	*p* < 0.0004 *
Progressive motility (%)	68.41 ± 24.57	72.00 (54.00–90.00)	31.73 ± 19.90	30.50 (17.25–42.75)	5.441 ^t^	*p* < 0.0001 *

^z^: Mann Whitney U test; t: Independent-samples *t*-test;. Q1: 25. percentile; Q3: 75. percentile; *: *p* < 0.01.

**Table 3 life-11-00933-t003:** Comparison of the proportions for embryo development according to the sperm selection methods. Data are given as mean ± SD.

Embryo Development	Microchip Group	Gradient Group	z	*p*-Value
1st day fertilized (n: 186)	0.89 ± 0.23 (n: 96)	0.91 ± 0.16 (n: 90)	−0.214	0.966
3rd day total (n: 170)	0.85 ± 0.26 (n: 89)	0.83 ± 0.25 (n: 81)	−0.761	0.570
3rd day G1 (n: 144)	0.83 ± 0.25 (n: 77)	0.76 ± 0.31 (n: 67)	−0.235	0.625
3rd day G2 (n: 26)	0.12 ± 0.23 (n: 12)	0.13 ± 0.21 (n: 14)	−2.452	0.933
5th day total (n: 109)	0.60 ± 0.31 (n: 62)	0.47 ± 0.30 (n: 47)	−0.585	0.050
5th day excellent (n: 63)	0.42 ± 0.28 (n: 41)	0.23 ± 0.23 (n: 22)	−1.037	0.029 *
5th day good (n: 18)	0.10 ± 0.14 (n: 10)	0.07 ± 0.12 (n: 8)	−1.497	0.471
5th day fair (n: 20)	0.02 ± 0.06 (n: 8)	0.01 ± 0.04 (n: 12)	−0.214	0.311
5th day poor (n: 8)	0.07 ± 0.11 (n: 3)	0.16 ± 0.19 (n: 5)	−0.761	0.104

z: Mann Whitney U test; * *p* < 0.05.

## Data Availability

Data is contained within the article or Appendix A.

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
