# Peer review of "Sperm Selection and Embryo Development: A Comparison of the Density Gradient Centrifugation and Microfluidic Chip Sperm Preparation Methods in Patients with Astheno-Teratozoospermia"

_life, 2021, doi:10.3390/life11090933_

Round 1

Reviewer 1 Report

This clinical study evaluated the usefulness of microfluidic methods in sperm selection. The employed  methods are established well and the story is sound. 

for the comments,
1)  Figure 1  photomicrograph
      the back-ground should be similar, so if possible fix it.

2) The fertilization rate and developmental rate were not different between two methods except high quality blastocyst. Besided, the author did not given the delivery rate.
Therefore, even though the microfluidic apparatus is available, we have to pay  to establish that system. So one of the good things is that in discussion, put some comments about that.

Author Response

1. “Figure 1 photomicrograph  the back-ground should be similar, so if possible fix it.”

We edited Figure 1 and corrected the background as suggested by the Reviewer.

2.  “The fertilization rate and developmental rate were not different between two methods except high quality blastocyst. Besided, the author did not given the delivery rate.

Therefore, even though the microfluidic apparatus is available, we have to pay to establish that system. So one of the good things is that in discussion, put some comments about that.”

We would like to thank the Reviewer for his / her valuable comments for the Discussion section. We underlined the requirement for further studies due to the mentioned reasons by the reviewer as “Additional research is required to explore the pregnancy and delivery rates” between lines 228-229.

We introduced the concerns about the costs as “Conversely, despite the continued evolvements in the technology, the system is not yet widely adopted in routine laboratory practice due to the higher costs in comparison to the centrifugation-based conventional techniques” between lines 223-225.

Reviewer 2 Report

In this manuscript, Çağla Güler et al. performed a study to compare the density gradient centrifugation and microfluidic chip sperm preparation techniques in Astheno-teratozoospermia patients. During the past two decades, the microfluidic chip technique has been well studied and demonstrated great potential as an alternative method for in vitro fertilization. However, its application in patients with astheno-teratozoospermia has not been well documented. In this study, the authors showed that the microfluidic technique could select sperm with higher motility comparing to traditional density gradient centrifugation methods. Importantly, sperm isolated using a microfluidic chip could generate higher quality blastocysts. Overall, this is a well-conducted study with sufficient evidence to support its conclusion and should be published.

Author Response

“In this study, the authors showed that the microfluidic technique could select sperm with higher motility comparing to traditional density gradient centrifugation methods. Importantly, sperm isolated using a microfluidic chip could generate higher quality blastocysts. Overall, this is a well-conducted study with sufficient evidence to support its conclusion and should be published.”

We would like to thank the Reviewer very much for his / her considerations.

Reviewer 3 Report

Life                                                       

COMMENTS TO THE AUTHOR

Manuscript ID 1356858: “ Sperm Selection and Embryo Development: Comparison of the Density Gradient Centrifugation and Microfluidic Chip Sperm Preparation Methods in Patients with Astheno-teratozoospermia

Dear Authors,

Please find enclosed the comments for the above-mentioned manuscript.

A SUMMARY OF THE CONTENT

The authors stated that the prospective study aims to compare the effects of the density gradient centrifugation and microfluidic chip sperm preparation methods on embryo development in a patient population with astheno-teratozoospermia. Selected spermatozoa prepared with either the microfluidic or density gradient methods were used to fertilize the mature sibling oocytes and semen parameters and embryo development on days 3 and 5 were assessed. The results showed that the density gradient group was associated with higher sperm concentration, while motility was significantly higher in the microfluidic chip group. There no significant differences in the fertilization rates and proportion of the 3rd day embryos. The proportions of the poor, fair and good blastocysts on day 5 were not significantly different, excellent blastocysts indicating high-quality embryos were observed in a significantly higher proportion in the microfluidic chip group. Authors concluded that the microfluidic chip sperm preparation yields sperm with higher motility and higher quality blastocysts at day 5 in patients with astheno-terato-zoospermia.

THE OVERALL OPINION OF THE MANUSCRIPT

The topic is attractive and could be of interest to the readers. However, the pioneered works as well as recent advances in the field published this year are not described in the introduction and in the discussion.

(1) INTRODUCTION

Please describe original, and important and pioneered results, as well as recent advance in the field. 

(2) DISCUSSION

Please describe original, and important and pioneered results, as well as recent advance in the field. 

(3) GENERAL

Please use official abbreviations.

Good luck and all the best :)

Author Response

“The topic is attractive and could be of interest to the readers. However, the pioneered works as well as recent advances in the field published this year are not described in the introduction and in the discussion.

(1) INTRODUCTION

Please describe original, and important and pioneered results, as well as recent advance in the field. “

We appreciate the Reviewer’s valuable comment regarding the Introduction section. We cited two more reviews with recent literature and the important studies in the field as “Microfluidic chip-based sperm sorting is a recent and powerful tool, which has been introduced as an alternative to centrifugation-based, conventional techniques [18-20]“ between lines 56-58.

19.Kashaninejad, N.; Shiddiky, M.J.A.; Nguyen, N.-T. Advances in Microfluidics-Based Assisted Reproductive Technology: From Sperm Sorter to Reproductive System-on-a-Chip. Advanced Biosystems 2018, 2, 1700197, doi:https://doi.org/10.1002/adbi.201700197.

20.Alias, A.B.; Huang, H.-Y.; Yao, D.-J. A Review on Microfluidics: An Aid to Assisted Reproductive Technology. Molecules 2021, 26, 4354.

Moreover, we introduced a phrase about the presence of several different microfluidic approaches as “A variety of microfluidic devices with different approaches ranging from passive to flow or chemical-based sorting have been designed to efficiently isolate the highly motile and healthy sperm [21-26].” between lines 58-59.

(2) DISCUSSION

Please describe original, and important and pioneered results, as well as recent advance in the field. “

The discussion section was improved by adding recent literature according to the Reviewer’s suggestion. Two more reviews added to the Introduction section were also cited in Discussion section as “In recent years, the microfluidic chip based sperm sorting has emerged as an alternative tool to the historical sperm preparation methods [18-20]” between lines 177-178. Furthermore, two recent publications were added to illustrate the recent results obtained with Fertile microfluidic chip as “However, a recent study did not demonstrate any differences in fertilization rate, embryo quality, blastocyst development, or pregnancy rate after sperm selection with microfluidic sorting or swim-up method [45]. On the other hand, a randomized controlled trial involving 181 patients with male factor infertility reported an enhanced pregnancy rate in the microfluidic group than that in the gradient group when the female age is above 35 and the total motile sperm count ranges between 1 and 5 million [46]” between lines 214-220.

(3) GENERAL

Please use official abbreviations.

As suggested by the reviewer, we removed MFC that was given as an abbreviation for microfluidic chip from the manuscript. We used the abbreviations such as ICSI, WHO and DGC.